# APOL1-Risk Genotype Induces Inflammatory and Hypoxic Gene Expression in Donor Kidneys

**DOI:** 10.3390/genes16091078

**Published:** 2025-09-15

**Authors:** Meghan Unes, Sree Kolli, Shaurya Mehta, Chandrashekhara Manithody, Jonathan Bruno, Krista L. Lentine, Ajay Jain, Mustafa Nazzal, Yasar Caliskan

**Affiliations:** 1School of Medicine, Saint Louis University, 1402 S. Grand Blvd, St. Louis, MO 63104, USA; meghan.unes@health.slu.edu (M.U.); sree.kolli@health.slu.edu (S.K.); 2Department of Pediatrics, School of Medicine, Saint Louis University, 1402 S. Grand Blvd, St. Louis, MO 63104, USA; shaurya.mehta@health.slu.edu (S.M.); chandrashekhara.manithody@health.slu.edu (C.M.); ajay.jain@health.slu.edu (A.J.); 3Division of Nephrology, SSM Health Saint Louis University Hospital, 1201 S. Grand Blvd, St. Louis, MO 63104, USA; jonathan.bruno@health.slu.edu (J.B.); krista.lentine@health.slu.edu (K.L.L.); 4Department of Surgery, SSM Health Saint Louis University Hospital, 1201 S. Grand Blvd, St. Louis, MO 63104, USA; mustafa.nazzal@health.slu.edu

**Keywords:** APOL1, normothermic machine perfusion, ischemia and reperfusion injury, hypoxia, kidney transplantation, graft outcomes, nephropathy

## Abstract

Background/Objectives: *APOL1* renal-risk variants (RRVs) are of increasing relevance to kidney disease and transplant outcomes. It is currently understood that the presence of RRVs in donors negatively impacts kidney allograft survival in an autosomal recessive pattern of inheritance. Less well known is the interplay between ischemia and alternative allograft preservation methods, such as normothermic machine perfusion (NMP), on APOL1 gene expression. To investigate this, we examined the effects of *APOL1* RRVs on APOL1 gene expression in ischemic donor kidneys and compared the differences in cytokine and APOL1 expression patterns between the alternative preservation methods, static cold storage (CS) and NMP. Methods: Non-utilized deceased donor kidney pairs from donors of African ancestry were procured from Mid-America Transplant after being deemed unsuitable for kidney transplant. Samples were collected from each donor kidney pair and DNA was extracted for *APOL1* genotyping. *APOL1* RRVs G1 (rs73885319) (rs60910145) and G2 (rs71785313) were identified by Sanger sequencing. From each pair, one kidney underwent 6 h NMP (n = 3) and the contralateral kidney 6 h of CS (n = 3) following the initial CS. Renal perfusion and biochemical, and histologic parameters were recorded. NMP was directly compared with CS using paired donor kidneys using NMP with allogeneic red blood cells, followed by assessment of perfusion, biochemical, and histologic parameters, in addition to gene expression. Results: Donor genotyping identified kidney pairs as heterozygous for the G1 RRV (G1/G0), homozygous for the G0 allele (G0/G0), and homozygous for the G2 RRV (G2/G2), respectively. All kidneys were successfully reperfused, with mRNA transcript levels of *APOL1*-related genes subsequently measured. Significant differences in *APOL1* gene expression were observed among all three groups of kidneys. In paired kidneys from baseline to hour 6 of NMP, mRNA expression varied significantly between G1/G0 and G2/G2 homozygous pairs (*p* = 0.002) as well as between the G0/G0 and G2/G2 pairs (*p* = 0.002). *APOL1* expression shifted by a significantly higher-fold change of 2.4 under NMP conditions in the G2/G2 genotype (*p* < 0.001). The inflammatory cytokine marker IFN-γ was also significantly upregulated in the G2/G2 genotype kidney, in both CS and NMP groups (*p* = 0.001). Other related genes such as KIM-1 were upregulated by a change of 3.9-fold in the NMP group for the G2/G2 kidney. Conclusion: Donor kidney pairs with the high-risk *APOL1* genotype, especially G2/G2, show increased *APOL1* expression and inflammation, particularly under NMP conditions. NMP enables detection of genotype-specific molecular changes in an ischemic reperfusion injury model, supporting its potential to improve donor kidney assessment before transplantation.

## 1. Introduction

### 1.1. APOL1 and Kidney Disease

Apolipoprotein 1 (*APOL1*) encodes a protein involved in high-density lipoprotein (HDL) metabolism and the innate immune system [1,2]. Its expression is upregulated by immunoregulatory signals [3]. Two coding variants, G1 and G2, are considered renal risk variants (RRVs), particularly prevalent in individuals of African ancestry [4]. Their involvement in *APOL1*-mediated kidney disease follows an autosomal recessive pattern of inheritance, meaning those heterozygous for either G1 or G2 variants (G1/G0 or G2/G0) do not carry an increased risk of developing kidney disease. On the contrary, individuals harboring two risk alleles (G1/G1, G2/G2, or G1/G2) possess an increased risk of kidney disease [2,5].

Individuals carrying two *APOL1* RRVs have nearly twice the risk of developing end-stage kidney disease (ESKD) compared to those without these alleles. In the United States, an estimated 13% of African American people carry two *APOL1* RRVs, elevating their susceptibility to kidney disease [6]. Interestingly, only a minority of risk allele carriers develop *APOL1*-mediated kidney disease [2,7,8,9]. This observation supports the hypothesis of a two-hit model requiring the presence of *APOL1* risk alleles and the contribution of environmental or systemic factors to initiate and promote *APOL1*-mediated kidney disease. Apart from the upregulated response to interferons, current evidence regarding the involvement of additional factors influencing APOL1 expression remains minimal.

### 1.2. APOL1 Expression and Its Implication in Kidney Transplantation

Development of *APOL1*-mediated kidney disease is a notable risk in transplant recipients receiving kidneys from donors with *APOL1* high RRVs [10]. Importantly, this risk appears to arise from the kidney itself, as the effects of *APOL1* high-risk variants are respective to the kidney and generated within the kidney cells. Podocytes, endothelial cells, and proximal tubular cells have been identified to express APOL1 [9,11]. Thus, any factors influencing *APOL1* expression in renal cells will subsequently influence the risk of developing *APOL1*-mediated kidney disease post-transplant.

Thus, donor *APOL1* status is of increasing relevance in kidney transplant evaluation [12]. Multiple studies have retrospectively demonstrated the increased risk of graft failure associated with the presence of 2 *APOL1* RRVs relative to 0 or 1 RRVs in donors [6,10,13]. This current understanding underlies the basis for our present aim, utilizing donor kidney pairs with different *APOL1* genotypes to investigate differences in expression patterns when under different storage conditions.

### 1.3. Organ Preservation Methods and Their Impact on APOL1 Gene Expression and Hypoxic Stress

Normothermic machine perfusion (NMP) is an alternative preservation method to static cold storage (SCS), which is receiving growing scientific interest owing to its clinical potential [14]. NMP delivers oxygenated blood or perfusate at physiologic temperatures to the kidney graft prior to transplantation. This preservation method allows for organ revitalization and improved functional assessment of solid organs prior to transplantation. By mimicking a physiologic state, NMP can potentially reduce the hypoxia associated with ischemia and reperfusion injury (IRI), in turn, promoting long-term graft function and mitigating the injurious complications related to SCS. In further contrast with SCS, which maintains organs in a hypometabolic state, NMP can allow the interrogation of dynamic molecular pathways, including immune activation and metabolic reprogramming. While potentially beneficial to mitigate complications of traditional cold storage (CS), this approach may alter the expression of injury-associated genes, including *APOL1*, and thereby influence graft viability, especially in kidneys with high-risk *APOL1* genotype. NMP also presents further opportunity in the form of organ assessment, which can be used to mechanistically screen for graft potential and transplant viability [15].

Hypoxia prior to implantation and IRI following perfusion can serve as environmental “second hits” that upregulate APOL1 expression in donor kidneys, thereby amplifying the cytotoxic effects of *APOL1* RRVs and increasing susceptibility to injury. Current evidence suggests that hypoxic disturbances stabilizing HIF-1α and subsequently inducing APOL1 expression are very important for furthering understanding of the utility of organ storage methods [16]. However, the interplay between hypoxia, HIF-1α and APOL1 regulation remains incompletely understood, particularly in the context of organ preservation. To address this gap, we examined how specific *APOL1* RRVs influence APOL1 expression and compared both cytokine and APOL1 expression patterns in kidney pairs preserved under the alternative preservation methods, CS and NMP. This pilot preclinical study seeks to further understanding of organ preservation methods in conjunction with donor genetic background to enhance transplantation strategies and improve graft outcomes.

## 2. Materials and Methods

### 2.1. Donors and Kidneys

Non-utilized kidney pairs from deceased donors of African ancestry were procured from Mid-America Transplant under a material transfer agreement with Saint Louis University, following Institutional Review Board (IRB) and Biosafety Committee approval (No. 2018-00040). All kidneys were deemed unsuitable for transplantation, and the IRB granted a waiver of donor consent. From each donor pair, one kidney underwent 6 h of NMP (n = 3) and the contralateral kidney underwent 6 h of CS (n = 3) following initial SCS. Perfusion and biochemical and histologic parameters were recorded and paired comparisons of NMP versus CS were conducted. In the NMP groups, kidneys were perfused with allogeneic red blood cells. Subsequent analyses included perfusion dynamics, biochemical parameters, histologic parameters, and gene expression profiling.

### 2.2. Donor Genotyping

Kidney samples were collected from three non-utilized deceased donor kidney pairs (n = 6). DNA extraction from kidney specimens was performed, and genotyping for *APOL1* RRVs G1 (rs73885319) (rs60910145) and G2 (rs71785313) was performed via Sanger sequencing.

### 2.3. RNA Extraction and Real-Time PCR Analysis

RNA extraction and real-time-PCR (RT-PCR) were performed at Saint Louis University to assess the impact of ischemia on renal *APOL1* expression. Total RNA was isolated from kidney tissue using the Allprep DNA/RNA mini kit (QIAGEN, Germany). cDNA was synthesized with the Verso cDNA Synthesis Kit (Thermo Fisher, Vilnius, Lithuania) and RT-PCR was carried out on a CFX Connect Real-Time Detection System (Bio-Rad, Hercules, California, USA) using iTaq SYBR Green Supermix (Bio-Rad, Hercules, California, USA). APOL1 expression levels were normalized to the housekeeping gene, β-actin. mRNA expressions of APOL1, HIF-1α, TGF-β, IFN-γ, and KIM-1 were analyzed at baseline (hour 0) and after 6 h of preservation under both CS and NMP. All assays were performed in triplicate. Gene expression profiles were compared between baseline and 6-h NMP samples.

### 2.4. Statistical Analysis

Statistical analyses were performed using SPSS for Windows (SPSS version 21.0, IBM Corp., Armonk, NY, USA) and R version 4.3.2 (31 October 2023). Descriptive statistics were summarized as frequencies with percentages for categorical variables and as medians with interquartile ranges (IQR) for continuous variables. Normally distributed data are presented as mean ± standard deviation (SD), and non-normally distributed data as median (IQR). Group comparisons were performed using t-tests or the Mann–Whitney U test, with the Welch t-test applied where appropriate. Associations between variables were assessed with Pearson’s correlation. All statistical tests were two-sided and a *p* value of 0.05 or less was considered statistically significant.

## 3. Results

### 3.1. Donor Characteristics and APOL1 Genotyping

All three pairs of kidneys were obtained from donation after brain death (DBD) donors with high kidney donor profile index (KDPI) scores (86–100%). Donor genotyping showed that the first kidney pair (K1) was heterozygous for G1 RRV (G1/G0), the second pair (K2) was homozygous (G0/G0), and the third kidney pair (K3) was homozygous for G2 RRV (G2/G2) (Table 1). The kidneys were successfully reperfused, with improved renal blood flow and resistance over the course of perfusion, and evidence of urine output (mean 30 mL/h). Kidney 1 (G1/G0) had no prior hypothermic machine perfusion (HMP) and demonstrated 75 mL/h urine output with NMP. Kidney 2 (G0/G0) had undergone HMP, but had a 5 mL/h urine output during NMP. Kidney 3 (G2/G2) also had prior HMP and had 10 mL/h urine output under NMP. Characteristics of non-utilized donor kidney pairs are shown in Table 1.

### 3.2. APOL1 mRNA Expression Patterns

At baseline, no significant differences in APOL1 expression were detected among ischemic kidney pairs preserved by CS (n = 3). After 6 h NMP, however, transcriptomic analyses revealed marked *APOL1* genotype-specific variation in APOL1 expression patterns. In contrast, the corresponding kidneys maintained in CS for 6 h showed no significant difference in APOL1 expression (*p* = 0.89). Under NMP, the G2/G2 genotype kidney (K3) showed the greatest induction, with a 2.43-fold increase in APOL1 expression compared with both G1/G0 (K1) and G0/G0 (K2) kidneys (*p* < 0.001). Significant variant-specific differences were observed between K1 vs. K3 (*p* = 0.002) and K2 vs. K3 (*p* = 0.002), highlighting the heightened susceptibility of G2/G2 kidneys to NMP-associated upregulation of APOL1 (Figure 1).

### 3.3. mRNA Expression Patterns of Hypoxia- and APOL1-Related Genes

At baseline, no significant differences were observed in the expression of APOL1 and related genes (HIF-1α, TGF-β, IFN-γ, and KIM-1) in ischemic kidney pairs at CS (n = 3).

#### 3.3.1. HIF-1α Expression Patterns

Unlike the APOL1 expression results, which showed significant variant-specific differences in expression only under 6-h NMP, HIF-1α showed significant variant-specific differences under both the CS and NMP conditions, with K3 (G2/G2 variant) consistently demonstrating the strongest response. After 6 h of CS, all variants showed significant differences, with the greatest expression difference exhibited between K2 and K3. K2 shifted by a 7.8-fold change and K3 by a change of 10.5-fold (*p* = 0.006). Additional significant differences in HIF-1α were observed between K1 and K2 (*p* = 0.037) and K1 and K3 (*p* = 0.013) under 6-h CS. Specifically, the K1 kidneys exhibited higher HIF-1α expression compared to K2, while K3 showed a marked upregulation in HIF-1α expression relative to K1 undergoing 6 h of CS. Under 6-h NMP, K3 demonstrated the greatest increase in HIF-1α expression, with an increase of 5.2-fold, compared with K1 (3.1, *p* = 0.027) and K2 (1.2, *p* = 0.009), highlighting the consistent, variant-specific responsiveness of G2/G2 kidney across preservation methods (Figure 2).

#### 3.3.2. IFN-γ Expression Patterns

Compared to APOL1 and HIF-1α, IFN-γ expression demonstrated the most robust overall response under both CS and NMP conditions. After 6 h NMP, all three kidneys showed highly significant increases in IFN-γ expression with changes of 29.3-fold (K1), 8.1-fold (K2), and 35.1-fold (K3). Risk variant-specific differences in IFN-γ expression following 6-h CS included K3 being dramatically higher than K1 (*p* < 0.001), as well as K2 (*p* < 0.001). Similarly, under 6 h NMP, IFN-γ expression in K3 was significantly higher than both K1 (*p* < 0.001) and K2 (*p* < 0.001) (Figure 3).

#### 3.3.3. KIM-1 Expression Pattern

mRNA expression results for KIM-1 showed no significant differences between variants at baseline or following 6-h CS. However, expression results amongst kidney pairs became significant following 6-h NMP. Most drastically, KIM-1 expression demonstrated significant differences between K2 and K3 (*p* < 0.001). Statistically significant differences in KIM-1 expression between K1 and K3 (*p* = 0.001) and K1 and K2 (*p* = 0.003) were also seen (Figure 4).

#### 3.3.4. TGF-β Expression Pattern

Amongst all genes analyzed, TGF-β showed the most significant *APOL1*-RRV-specific upregulation, with K3 demonstrating marked sensitivity to both 6-h CS and NMP. The strongest overall increase was observed under 6-h NMP, where K3 exhibited a 24.7-fold increase (*p* < 0.001). In contrast, K2 showed the weakest TGF-β upregulation under 6-h NMP, with a modest, but statistically significant 1.9-fold increase (*p* < 0.001) (Figure 5).

The RT-PCR results, showing relative mRNA gene expression profiles of *APOL1* and associated genes, presented as fold change, are illustrated in Figure 1, Figure 2, Figure 3, Figure 4 and Figure 5.

## 4. Discussion

### 4.1. Genotype-Specific APOL1 Responses to Hypoxia and Injury

This study demonstrated *APOL1* genotype-specific differences in APOL1 expression across storage conditions. During 6-h NMP, APOL1 expression was 2.43-fold higher in G2/G2 kidneys compared to that in G1/G0 and G0/G0. The G2/G2 genotype also exhibited marked upregulation of IFN-γ under both CS and NMP, along with a 3.9-fold increase in KIM-1 during NMP, indicative of substantial tubular injury. Together, these findings highlight the link between *APOL1* risk variants, hypoxia, and injury in a non-utilized donor kidney perfusion model.

High-risk *APOL1* genotypes (G1/G1, G1/G2, and G2/G2), particularly prevalent in individuals of African ancestry, confer increased susceptibility to kidney injury via a toxic gain-of-function effect [16]. APOL1 expression in renal cells, especially podocytes and tubular cells, is upregulated by HIF-1α and IFN-γ. HIF-1α, stabilized under hypoxic conditions, promotes APOL1 transcription by binding to hypoxia response elements in the APOL1 enhancer, a process augmented by IFI16 [16]. IFN-γ induces APOL1 expression through the JAK–STAT and IRF pathways and acts synergistically with HIF-1α to further amplify transcription. However, only a minority of individuals of African ancestry carrying *APOL1* RRVs develop kidney disease, suggesting that additional genetic or environmental modifiers are involved. Both HIF-1α and IFN-γ expression are regulated by distinct mechanisms, and functional polymorphisms in these genes may alter mRNA or cytokine production, thereby influencing APOL1 expression. Likewise, polymorphisms in TGF-β and KIM-1 may modulate downstream injury responses to APOL1. Incorporating polygenic risk analyses of HIF-1α, IFN-γ, TGF-β, and KIM-1 could help refine the interpretation of experimental findings and clarify the broader genetic landscape influencing *APOL1*-mediated kidney injury.

In the present study, TGF-β, a key mediator of renal fibrosis, was also markedly upregulated in G2/G2 kidneys, particularly under NMP, though its direct mechanistic link to APOL1 remains less defined [17]. It likely acts downstream or in parallel, contributing to glomerulosclerosis. KIM-1, a sensitive marker of tubular injury, reflects the extent of cellular stress and injury in high-risk *APOL1*-genotype carriers, especially under hypoxic and inflammatory conditions [17,18]. Together, these results highlight the interplay between *APOL1* RRVs, hypoxia, inflammation, fibrosis, and tubular injury, all of which are key drivers of the heterogeneity and severity seen in *APOL1*-mediated nephropathy.

### 4.2. Impact of Kidney Preservation Methods on APOL1 and Cytokine Expression

The significant upregulation in HIF-1α expression in both CS and NMP conditions across all donor kidney genotypes is consistent with the current understanding that hypoxic insult to kidney allografts results in HIF stabilization. Previous studies demonstrate that, in conjunction with HIF involvement in kidney diseases, hypoxic insults, such as those observed in disease-induced acute kidney injury (AKI) or sickle cell disease, drive the progression of kidney disease by promoting an increase in APOL1 expression [19,20]. Our findings demonstrate that the kidney storage method profoundly influences APOL1 expression in an *APOL1* genotype-dependent manner. While APOL1 expression levels slightly increased across *APOL1* RRV genotypes at baseline and after 6 h of CS, NMP revealed marked genotype-specific effects with G2/G2 kidneys, showing a greater than 2-fold induction compared to G1/G0 and G0/G0. This likely reflects the physiologic and metabolic milieu created by NMP, including restored oxygenation, hemodynamic stress, and activation of inflammatory pathways, which can serve as a “second hit” to upregulate APOL1 in high-risk genotypes. Mechanistically, CS suppresses global transcription, including genes involved in energy generation and inflammatory pathways, whereas NMP activates oxidative phosphorylation and immune/inflammatory responses. HIF-1α and IFN-γ, both induced during NMP, are potent transcriptional drivers of APOL1 in podocytes and tubular cells, further explaining the heightened response in G2/G2 kidneys. Clinically, these findings may highlight the need to consider genetic background when evaluating preservation strategies. NMP may inadvertently amplify APOL1 expression in high-risk grafts, potentially promoting downstream inflammatory and cytotoxic pathways, while CS maintains a hypometabolic state that limits transcriptional activation. Although NMP improves overall organ preservation and functional recovery, its effect on APOL1 underscores the potential value of targeted interventions, including cytokine modulation or APOL1 pathway inhibition during NMP. Future studies in larger donor cohorts, incorporating functional injury assays and long-term graft outcomes, are necessary to validate these observations and assess whether genotype-tailored preservation strategies can mitigate the high genotype-associated risk of *APOL1*.

### 4.3. APOL1-Mediated Kidney Injury and Graft Outcomes

The *APOL1* high-risk genotype increases the likelihood of kidney injury and diminished graft function, particularly in individuals of African ancestry. In the assessment of the prognosis of *de novo* post-kidney transplant collapsing focal segmental glomerulosclerosis (cFSGS), Santoriello et al. demonstrated that recipients of kidneys from donors with *APOL1* high-risk genotypes independently predicted poorer graft outcomes relative to grafts harboring *APOL1* low-risk genotypes [21]. These high-risk genotypes may increase susceptibility to IRI, which is a critical determinant in early graft function and significantly disrupts the natural physiology of kidney allografts [22,23,24,25,26].

mRNA expression results from our present study suggest a dramatic upregulation in APOL1 expression in the *APOL1* G2/G2 kidney that underwent NMP, which in turn, may have exacerbated this pathophysiology. With respect to allograft outcomes, previous studies show that donor kidneys with two *APOL1* RRVs have significantly decreased graft survival in situations of prolonged ischemia. Thus, these studies provide further support that grafts harboring two *APOL1* RRVs relative to 0 or 1 variants are stronger determinants in shorter kidney allograft survival [10,27,28]. While kidney pairs used in our study were non-utilized human kidneys and not used in subsequent kidney transplantation, future studies are important to elaborate upon the variant-specific findings our results point to.

### 4.4. APOL1 Genotype and Its Emerging Importance in Transplant Evaluation and Outcomes

In addition to the current literature demonstrating decreased graft survival time for donors with two *APOL1* RRVs in prolonged ischemic conditions, our study further suggests that *APOL1* status may play a valuable role in pre-transplant evaluation and donor organ risk stratification. It is equally important to appreciate the role genetic testing plays in transplant planning without exacerbating disparities in access and clinical care. Thus, *APOL1* status and genotype-specific conclusions about the impact of high-risk variants on transplant outcomes are of increasing importance but must be investigated and appreciated carefully.

Emerging strategies in organ preservation methods, such as NMP, offer strong potential for the assessment of genotype-specific injury responses as well as an opportunity to inform targeted intervention strategies for transplant recipients of high-risk donor kidneys.

### 4.5. Study Limitations

Limitations of this study include the pilot nature, as we report here on findings from three, non-utilized deceased-donor kidney pairs. This small sample size reduces generalizability and statistical power, and further studies should be conducted to determine true causation. Secondly, increasing the *APOL1* allele frequencies amidst a larger sample size will strengthen genotype-specific conclusions about the different *APOL1* genotypes and HIF-1α interactions on *APOL1* gene expression in donor kidney preservation methods. Third, there is variability in donor characteristics such as ischemia time, age, and KDPI on gene expression. Given the characteristics of three donors, stratification and understanding of these variables as potential confounders is limited. Lastly, our data focuses on *APOL1* gene expression rather than protein levels. Further studies correlating expression data with protein secretion can further contribute to pathophysiologic understanding. While our analysis provides preliminary insight demonstrating increased APOL1 expression under NMP conditions, larger cohorts are needed to validate this interplay. Thus, our results serve to drive further hypotheses and larger cohort validation.

## 5. Conclusions

In this pilot preclinical study using non-utilized deceased-donor kidney pairs from donors of African ancestry, kidneys with *APOL1* G0/G0, G1/G0, and G2/G2 risk variants demonstrated different transcriptomic responses to alternative preservation methods (traditional CS and NMP). These preliminary findings may contribute to ongoing intervention strategies for improving transplantation outcomes and suggest the clinical importance of targeted therapeutics for managing *APOL1-*mediated kidney diseases, particularly for patients carrying the high-risk *APOL1* variants as well as allografts harboring high-risk *APOL1* variants. Further studies are needed to solidify the interplay between *APOL1*, inflammation, and kidney injury in the setting of allograft ischemia and reperfusion.

## Figures and Tables

**Figure 1 genes-16-01078-f001:**
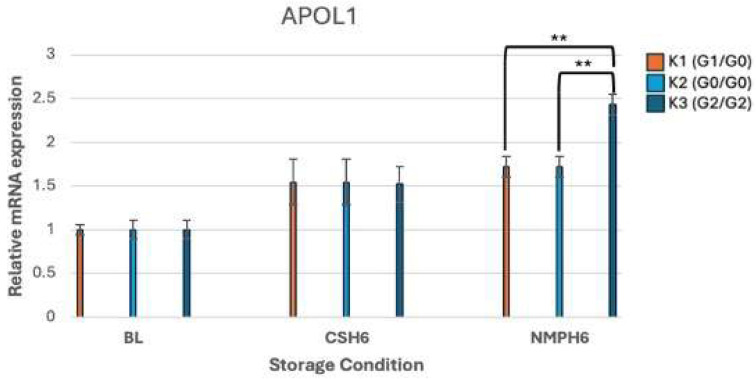
mRNA expression levels of APOL1. ****** *p* < 0.01.

**Figure 2 genes-16-01078-f002:**
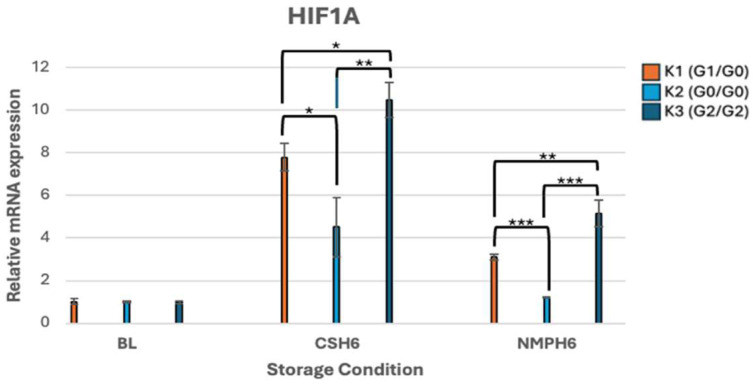
mRNA expression levels of HIF-1α. ***** *p* < 0.05, ****** *p* < 0.01, *******
*p* <0.001.

**Figure 3 genes-16-01078-f003:**
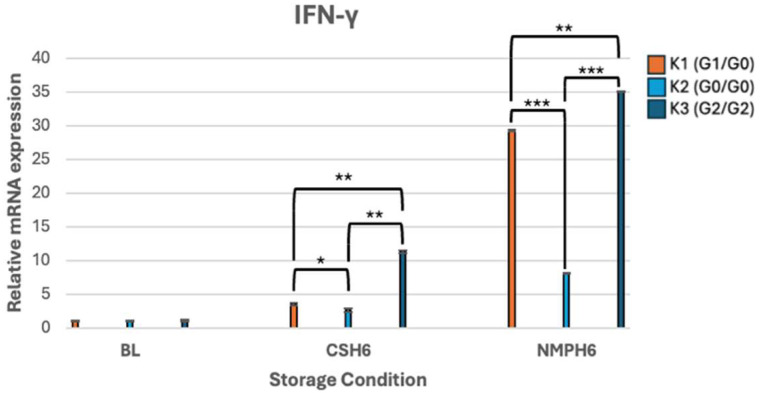
mRNA expression levels of IFN-γ. ***** *p* < 0.05, ****** *p* < 0.01, *******
*p* <0.001.

**Figure 4 genes-16-01078-f004:**
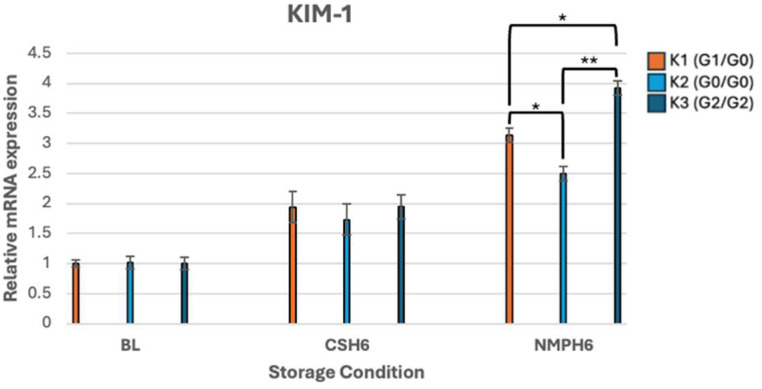
mRNA expression levels of KIM-1. ***** *p* < 0.05, ****** *p* < 0.01.

**Figure 5 genes-16-01078-f005:**
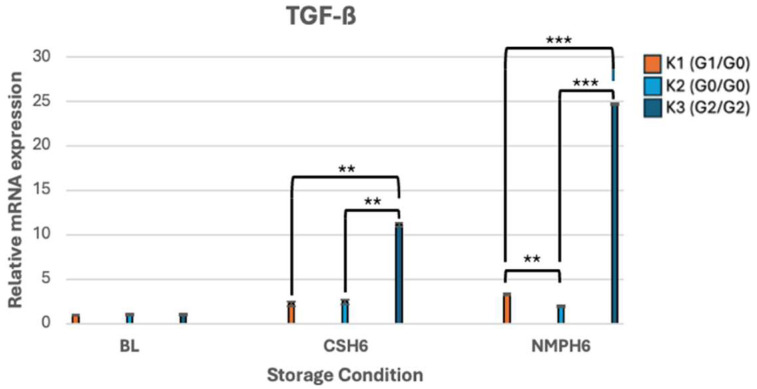
mRNA expression levels of TGF-β. ******
*p* < 0.01, *******
*p* < 0.001.

**Table 1 genes-16-01078-t001:** Characteristics of Non-Utilized Deceased Donor Kidney Pairs (n = 3).

	Kidney 1	Kidney 1	Kidney 2	Kidney 2	Kidney 3	Kidney 3
*APOL1* genotype	G1/G0	G1/G0	G0/G0	G0/G0	G2/G2	G2/G2
Preservation method	CS	NMP	CS	NMP	CS	NMP
Donor type	DBD	DBD	DBD	DBD	DBD	DBD
Age (yo)	70	70	47	47	64	64
KDPI (%)	100	100	86	86	96	96
CIT (hours)	17	17	31	31	26	26
Previous HMP	-	-	+	+	+	+
Urine output (mL/hour)	0	75	0	5	0	10

Abbreviations: CIT: cold ischemia time, CS: cold storage, DBD: donation after brain death, HMP: hypothermic machine perfusion, NMP: normothermic machine perfusion, KDPI: kidney donor profile index.

## Data Availability

The data presented in this study are available on request from the corresponding author.

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
