# Peer review of "APOL1-Risk Genotype Induces Inflammatory and Hypoxic Gene Expression in Donor Kidneys"

_genes, 2025, doi:10.3390/genes16091078_

Round 1

Reviewer 1 Report

Comments and Suggestions for Authors

MS “APOL1 Risk Genotype Induces Inflammatory and Hypoxic Gene Expression in Donor Kidneys” by Unes et al., submitted to Genes, compares two preservation methods: 1) normothermic machine perfusion (NMP) and 2) static cold storage (SCS). The compared donor kidneys were from donors with compound heterozygous (G1/G2) or homozygous (G1/G1 or G2/G2) APOL1 risk genotypes and G0/G0 donors. While the scope of this manuscript is valuable, the study design with no more than n = 2 kidneys per genotype does not permit statistical evaluation. Another methodological flaw is the focus on APOL1 expression rather than protein secretion, which appears to be more relevant to the transplant’s impact on host immunity.

In addition, the manuscript is difficult to follow because it lacks structured sections in the Introduction describing transplantation; the rationale for using organs with gene alleles that aberrantly influence kidney function; the frequency of APOL1 risk alleles in transplants; and the mechanisms of their pathogenicity.

Comments on the Quality of English Language

MS is not structured into logical sections. For example, although the MS addresses storage of transplanted kidneys, it starts the Introduction with: “Apolipoprotein 1 (APOL1) is a gene that plays a crucial role in high-density lipoprotein (HDL) particles in the blood and is also involved in the innate immune system [1,2].” Similarly disorganized are the Abstract, Results, and Discussion sections.

Author Response

Comment #1: While the scope of this manuscript is valuable, the study design with no more than n = 2 kidneys per genotype does not permit statistical evaluation. Another methodological flaw is the focus on APOL1 expression rather than protein secretion, which appears to be more relevant to the transplant’s impact on host immunity.

Reply: We thank you for this notable point. We agree with your interpretation of our present data and note that this limitation is mentioned in our Study Limitations section (4.5). We note the small sample size limiting generalizability and emphasize that our paper provides “preliminary insights” serving to “drive further hypothesis” and “larger cohort validation.” Additionally, we added your critique regarding expression vs protein secretion to our study limitations stating that, “our data focuses on APOL1 gene expression rather than protein levels. Further studies correlating expression data with protein secretion can further contribute to pathophysiologic understanding.” We thank you again for this point.

Comment #2: In addition, the manuscript is difficult to follow because it lacks structured sections in the Introduction describing transplantation; the rationale for using organs with gene alleles that aberrantly influence kidney function; the frequency of APOL1 risk alleles in transplants; and the mechanisms of their pathogenicity.

Reply:

We agree with your suggestion on structuring. To provide an organized introduction to our study, we included the following subheadings in the introduction section:

1.1 APOL1 and Kidney Disease

            In this section, we discuss the pathogenic mechanism of APOL-1 mediated kidney disease per the reviewers’ comment. This is the 1st introduction section as it forms the basis for reader understanding of APOL1 mediated kidney disease and relevance in transplant and organ storage methods. We also address APOL1 high risk variants associated with increased risk of developing ESKD compared to those without the risk alleles.

1.2 APOL1 Expression and its Implication in Kidney Transplantation

            In this section, we address the important points of (1) frequency of APOL1 risk allele in transplants, and the study rationale for using organs with high-risk gene alleles. We present why these notable points underlie our present aim to assess donor kidneys harboring different APOL1 genotypes to assess differences in expression patterns under different storage conditions.

1.3 Organ Storage Methods and Their Impact on APOL1 Gene Expression and Hypoxic Stress

            In this section, we discuss the clinical potential of NMP and make notable additions including our reasoning behind assessing the interplay between hypoxia, HIF-1a expression, and APOL1 regulation in the context of organ preservation methods.

Comment #3: MS is not structured into logical sections. For example, although the MS addresses storage of transplanted kidneys, it starts the Introduction with: “Apolipoprotein 1 (APOL1) is a gene that plays a crucial role in high-density lipoprotein (HDL) particles in the blood and is also involved in the innate immune system [1,2].” Similarly disorganized are the Abstract, Results, and Discussion

Reply:

Thank you for the opportunity to revise based upon this critique. The abstract was revised to flow better with our study rationale. The results section was elaborated upon by making more clear mention of donor genotyping and characteristics of donor kidneys as well as simplification of the presentation of gene expression patterns. All additions/revisions are highlighted. The discussion section was re-structured into the following sections: 4.1 Genotype Specific APOL1 Responses to Hypoxia and Injury, 4.2 Impact of Kidney Storage Methods on APOL1 and Cytokine Expression, 4.3 APOL1-Mediated Kidney Injury and Graft Outcomes, 4.4 APOL1 Genotype and its Emerging Importance in Transplant Evaluation and Outcomes, 4.5 Study Limitations. All additions are highlighted, paying particular attention to reviewer critiques about the discussion of potential contributors from genetic polymorphisms and storage method effects on APOL1 and cytokine expression.

Reviewer 2 Report

Comments and Suggestions for Authors

Paper by Unes et al. analyzes the relationship among APOL1 polymorphisms and  expression of some cytokine and APOL1 genes in human kidneys preserved with  static cold or normothermic machine perfusion storage methods.

Experiments reported in the manuscript explore a relevant topic as the presence of risk related  APOL1 polymorphisms, when expressed in homozygosis or in double heterozygosis, might be associated to renal diseases, which may occur in the transplanted kidney.  However some points of the manuscript necessitate to be clarified.

Line 100 Authors reports that they performed experiments on three “non-utilized deceased donor kidneys” however, as better reported in the abstract, and considering  data reported in the results, authors refer to three non-utilized deceased donor kidney pairs. The authors must disambiguate this point in materials and methods and in following result description.

“Materials and methods” need to be revised and divided into more sections. i.e., Donors and kidneys (in this section, donors' consent to the use of organs for research purposes should be more clearly stated), Gene typing, RNA extraction and real-time PCR analysis, Statistical analysis.

In the discussion authors briefly stated that “HIF-1α, stabilized under hypoxic conditions, promotes APOL1 transcription by binding to hypoxia response elements in the APOL1 enhancer, a process augmented by IFI16[11]. IFN-γ induces APOL1 via the JAK-STAT and IRF pathways and acts synergistically with HIF-1α to further amplify expression.”

This is an interesting point. In principle, should be considered that only a minority of Blacks bearing APOL1 risk variants develop kidney disease. This suggests that other genetic or environmental factors are involved. As well known, both HIF-1α and IFN-γ expression are influenced by different mechanisms. In particular genetic polymorphisms of these gene might impinge on mRNA and cytokine productions. So it is hypothesizing that APOL1 expression might be affected in turn by functional polymorphisms of HIF-1α and IFN-γ. Similarly  response to APOL1 might be influenced by polymorphisms of TGF-β and KIM-1.

In this view, appear mandatory that  HIF-1α, IFN-γ, TGF-β and KIM-1 genetic polymorphisms should be assessed to avoid mis-interpretations of the results reported in the paper.

Discussion on the effect of kidney storage methods on APOL1 and cytokine expressions is missed. Considering that paper is based on comparison of RNA expression among CS and NMP stored kidneys, a discussion chapter devoted to this topic should be added.

Author Response

Comment #1: Line 100 Authors reports that they performed experiments on three “non-utilized deceased donor kidneys” however, as better reported in the abstract, and considering data reported in the results, authors refer to three non-utilized deceased donor kidney pairs. The authors must disambiguate this point in materials and methods and in the following result description.

Reply: We thank you for pointing out this need for clarification. To eliminate ambiguity, all mention of donor kidneys is stated with the following phrasing: “non-utilized deceased donor kidney pairs.” We additionally provided further clarity in Section 2.1 stating that, “From each donor pair, one kidney underwent 6 hours of NMP (n=3) and the contralateral kidney underwent 6 hours of CS (n=3) following initial SCS. We thank you again for the opportunity to provide this clarity.

Comment #2: “Materials and methods” need to be revised and divided into more sections. i.e., Donors and kidneys (in this section, donors' consent to the use of organs for research purposes should be more clearly stated), Gene typing, RNA extraction and real-time PCR analysis, Statistical analysis.

Reply: We appreciate this critique. To provide further clarity to study design, we reorganized the Materials and Methods section with the following subheadings and subsequent elaborations: 2.1 Donors and Kidneys; 2.2 Donor Genotyping; 2.3 RNA extraction and real-time PCR Analysis; 2.4 Statistical Analysis

Comment #3: In the discussion authors briefly stated that “HIF-1α, stabilized under hypoxic conditions, promotes APOL1 transcription by binding to hypoxia response elements in the APOL1 enhancer, a process augmented by IFI16[11]. IFN-γ induces APOL1 via the JAK-STAT and IRF pathways and acts synergistically with HIF-1α to further amplify expression.” This is an interesting point. In principle, should be considered that only a minority of Blacks bearing APOL1 risk variants develop kidney disease. This suggests that other genetic or environmental factors are involved. As well known, both HIF-1α and IFN-γ expression are influenced by different mechanisms. In particular genetic polymorphisms of these gene might impinge on mRNA and cytokine productions. So it is hypothesizing that APOL1 expression might be affected in turn by functional polymorphisms of HIF-1α and IFN-γ. Similarly, response to APOL1 might be influenced by polymorphisms of TGF-β and KIM-1.In this view, appear mandatory that HIF-1α, IFN-γ, TGF-β and KIM-1 genetic polymorphisms should be assessed to avoid mis-interpretations of the results reported in the paper.

Reply: Thank you for this notable point. To strengthen our discussion and avoid mis-interpretation of results, we revised discussion section 4.1 (Genotype-Specific APOL1 Responses to Hypoxia and Injury) to incorporate discussion of potential polymorphisms modulating downstream responses to APOL1. These additions are highlighted and included with original text explaining the mechanisms behind the genes assessed and the reasons for using these genes as markers for cellular injury.

Comment #4: Discussion on the effect of kidney storage methods on APOL1 and cytokine expressions is missed. Considering that paper is based on comparison of RNA expression among CS and NMP stored kidneys, a discussion chapter devoted to this topic should be added.

Reply: Thank you for this critique. To strengthen this aspect of our paper, we elaborated upon discussion of both NMP and CS storage methods and their impact on gene expression as seen in highlighted additions to the text in discussion section 4.2.

Round 2

Reviewer 2 Report

Comments and Suggestions for Authors

Paper by Unes et al. has been re-managed according to the suggestions. Answers to my questions are satisfactory. So, in my opinion, paper is now suitable for publication on Genes.